# Prevalence of HIV, risk behaviours and vulnerabilities of female sex partners of the HIV positive people who inject drugs (PWID) in Dhaka city, Bangladesh

**Md. Masud Reza[1], A. K. M. Masud Rana[1], Mohammad Niaz Morshed Khan[1], Md. Safiullah Sarker[1], Sujan Chowdhury[1], Md. Ziya Uddin[2], Lima Rahman[3], Mohammad Ezazul Islam Chowdhury[4], Md. Abu Taher[5], Sharful Islam Khan[1]***

1 Programme for HIV and AIDS, Infectious Disease Division, International Centre for Diarrhoeal Disease Research, Bangladesh (icddr,b), Dhaka, Bangladesh, 2 HIV/AIDS Specialist, UNICEF, Dhaka, Bangladesh, 3 Director, Health and Nutrition Sector, Save the Children in Bangladesh, Dhaka, Bangladesh, 4 Technical Advisor, HIV/AIDS Program, Save the Children in Bangladesh, Dhaka, Bangladesh, 5 National Programme Coordinator (Drugs and HIV/AIDS), UNODC, Dhaka, Bangladesh

* sharful@icddrb.org

**Data Availability Statement:** The data that we have used in this manuscript are not publicly available due to data confidentiality, ethical

## Abstract

### Background

The HIV epidemic in Bangladesh is largely being driven by people who inject drugs (PWID) and mainly concentrated in Dhaka city. Intregrated biological and behavioural survey (IBBS) data of 2016 showed that a considerable percentage of the HIV positive PWID had unsafe sex with their female sex partners. Prevalence of HIV, risk behaviorus and vulnerabilities among the female sex partners of the PWID still remain unexplored.

### Methods

To measure HIV prevalence, risk behaviours (drugs/injection/sexual) and vulnerabilities (treatment of and knowledge of sexually transmitted infections (STIs) and HIV/uptake of the routine HIV testing and HIV prevention services/physical and sexual violence), a quantitative survey was conducted among 227 female sex partners of the HIV positive PWID in Dhaka city in 2019 by adopting a take-all sampling technique.

### Results

The median age of participants was 34.0 years. Prevalence of HIV was 16.7% (95% CI: 12.4–22.2). Only 6.8% to 18.7% of the participants used condoms consistently with different male sex partners; only 6.8%cto 18.7% during last year. Seventy five percent (95% CI: 69.2–80.8) had no knowledge on STI symptoms. Self-reported symptoms of STIs were reported by 26% (95% CI: 20.7–32.1) and half sought treatment during last year. Nineteen percent (95% CI: 14.7–25.1) had comprehensive knowledge of HIV. As part of the routine HIV prevention services by the PWID drop-in-centres (DICs), 42.7% (95% CI: 36.4–49.3) of the participants were tested for HIV and knew their result within the last year. One-third

restrictions and data policy of icddr,b. However, data are available from Ms. Armana Ahmed (armana@icddrb.org), Head of Research Administration of icddr,b upon reasonable request.

**Funding:** funded by The Global Fund to Fight AIDS, Tuberculosis and Malaria (GFATM), through the Grant 'Expanding HIV/AIDS Prevention in Bangladesh' under the terms of Grant Agreement NO. GR-01603 with icddr,b. icddr,b acknowledges with gratitude the commitment of the Global Fund to its research efforts. icddr,b is also thankful to the Governments of Bangladesh, Australia, Canada, Sweden and the UK for providing core/unrestricted support to icddr,b.

**Competing interests:** The authors do not have any competing interests.

never received HIV prevention services. During the last one year preceding the survey, 46% (95% CI: 39.3–52.6) reported been beaten and 20.2% (95% CI: 15.3–26.1) been raped.

## Conclusion

It is urgently necessary to consider the high-risk behaviours and vulnerabilities in designing or to strengthen targeted interventions for female sex partners of the HIV positive PWID in Dhaka city to ensure equality in accessing and utilization of services.

## Introduction

The epidemic of HIV in Bangladesh is largely being driven by the people who inject drugs (PWID) and is mainly concentrated in Dhaka city. According to the data of the HIV surveillance from 2016, the prevalence of HIV among the PWID in Dhaka city has risen to an alarming situation. In Dhaka city, the prevalence of HIV significantly increased to 22.0% in 2016 from 5.3% in 2011 [1]. Data from the 2016 HIV surveillance also showed that among those PWID who were HIV positive, approximately 30% were married. A considerable percentage (35%-84%) of the HIV positive PWID had sex with wives/female sex workers (FSWs)/non-transactional female sex partners (girlfriends/relatives/neighbours) in the last year. Condom use with female sex partners was very low; only 25%-40% PWID reported that condoms were used consistently in all sex acts in the last one year prior to the HIV surveillance in 2016. This data highlights that female sex partners of the HIV positive PWID in Dhaka city are at great risk of HIV infection.

In Bangladesh, there is no data on the prevalence of HIV, risk behaviours and vulnerabilities among females who have sex particularly with the HIV positive PWID, and no comprehensive HIV prevention services in place for them. Therefore, to design or strengthen HIV prevention services for female sex partners of the HIV positive PWID, a cross-sectional study was conducted in Dhaka city in 2019 using mixed methods (qualitative and quantitative) approaches in 2019, to measure the prevalence of HIV, risk behaviours, vulnerabilities and collect information about the availability of HIV prevention services for these population groups.

## Research design and methods

### Study type

This study was conducted following a cross-sectional design using mixed methods (qualitative and quantitative) approach among female sex partners of the HIV positive PWID in Dhaka city.

### Study population

Female sex partners were categorised into three groups who are referred to as study participants in this study, such as Group-1: wives, Group-2: FSWs, and Group-3: non-transactional/other female sex partners (except wives and FSWs such as girl-friends/relatives/neighbours) of the HIV positive PWID who receive services from 14 comprehensive drop-in-centres (CDICs)/drop-in-centres (DICs) in Dhaka city. The inclusion criteria for the participants were those who were wives of the HIV positive PWID and those who were FSWs or other female sex partners and had sex with a HIV positive PWID in the last one year prior to the survey

and, willing to provide informed written consent/assent to test HIV and to provide risk behavioural information.

## Sample size

Before calculating the sample size for the quantitative survey, we performed a literature review to find out the prevalence of HIV among female sex partners of the PWID in other countries. We found nine studies. The countries include: India [2, 3], Pakistan [4], Vietnam [5], Kazakhstan [6], Iran [7], Kyrgyzstan [8], Russia [9] and Malaysia [10]. The data showed that the prevalence of HIV among the female sex partners of PWID ranges from 2.5%-45.0% in these countries, thus indicating the wide variation of estimates. Using the standard formula mentioned below and the prevalence of HIV among the female sex partners of PWID from the above-mentioned countries, the calculated sample sizes ranged between 37–380. At the same time, there is no such data in Bangladesh. However, in Bangladesh we had data on the prevalence of HIV among PWID from the IBBS that was conducted in Dhaka in 2016 [1]. Due to the lack of required data in Bangladesh, we assumed that the prevalence of HIV among PWID would be equal to the prevalence of HIV among their female sex partners. Then, in order to conduct a cross-sectional study among female sex partners of the HIV positive PWID, we calculated three indicators from the IBBS dataset using Stata that are described below.

The indicators and the estimated percentage values were as follows:

1. Indicator for Group-1 (wives): Prevalence of HIV among those male PWID who were currently married = 20.7%

2. Indicator for Group-2 (FSWs): Prevalence of HIV among those male PWID who bought sex from FSWs in the last 12 months = 28.8%

3. Indicator for Group-3 (girlfriends/relatives/neighbours): Prevalence of HIV among those male PWID who had sex with non-transactional females (except FSWs and wives) in the last 12 months = 21.3%

Thereafter, to calculate the sample size for each of the groups mentioned above the following formula-1 [11] was used.

$$n_1 = \frac{z^2_{1-\frac{\alpha}{2}}}{d^2} pq$$

In the above equation:

$n_1$ = Calculated sample size

p = Percentage values of the indicators from four groups of the male PWID mentioned above

q = 1-p

$Z_{1-\alpha/2}$ = The Z-score corresponding to the desired level of significance = 1.96 (at the 95% confidence interval)

d = Desired level of precision = 5%

The sample size was calculated for each of the indicators and a total of 825 (252 for Group-1, 30.6% of the total sample size; 315 for Group-2, 38.2% of the total sample size; 258 for Group-3, 31.2% of the total sample size) was calculated. Since the total number of the HIV positive male PWID who were currently alive and enlisted in the DICs in Dhaka city while the proposal of this study was being prepared was already known beforehand, the total sample size was further adjusted for the finite population correction (FPC) according to the following

formula below [12], thus providing an achievable sample size.

$$n_2 = \frac{n_1}{1 + \frac{n_1}{N}}$$

In the above equation:

$n_2$ = Calculated sample size after FPC and adjusting for drop-outs during risk behaviour interview

N = Total number of the HIV positive male PWID live in Dhaka city = 619 [13]

After adjusting the total sample size with FPC, the total sample size was 354. Using the proportionate percentage points of sample sizes mentioned above, the sample size for group-1 became 108 wives, 135 for FSWs and 111 for other female sex partners. Finally, the sample size was inflated by 5% to adjust for drop out during risk behaviour interview that resulted in 114 wives in Group-1, 142 FSWs in Group-2 and 116 non-transactional female sex partners in Group-3. This amounted to a total target sample size of 372 female sex partners of the HIV-positive PWID for this study.

## Sampling

At the beginning of the study, to make a sampling frame of female sex partners, a mapping exercise of the HIV positive PWID was conducted in Dhaka city who were the only gatekeepers for accessing to the participants. They were asked whether or not they want to allow their female sex partners (wives, FSWs and others) to take part in this study. During this process, a total of 411 HIV positive PWID were contacted. Of them, 158 agreed that they will allow us to access to their wives, 69 of them agreed that they will allow us to access to FSWs and 9 agreed that they will allow us to access to non-transactional female sex partners. Thus, a total of 236 female sex partners of the HIV positive PWID were expected to be included in the study. Detailed addresses of the HIV positive PWID were listed, which constituted the sampling frame of female sex partners. Thereafter, the total number of female sex partners counted during mapping was compared with the target sample sizes. Since the number of female sex partners counted during mapping was smaller particularly for FSWs and other female sex partners, a take-all sampling approach was adopted to collect quantitative data for all groups. Finally, the quantitative survey was conducted among 227 participants adopting a take-all sampling technique to assess their risk behaviours, vulnerabilities and to determine HIV status. In the recruitment of participants, we have leveraged all of their network sources such as staff members from the CDIC/DIC, case workers (who provide services to the HIV positive PWID) and the HIV positive PWID.

## Process of quantitative data collection and HIV testing

Before approaching the participants for interview and HIV testing, prior written consent was taken from the HIV positive PWID to include their female sex partners in the study. Thereafter, informed written consent (those who were 18 years of above) or assent (those who were below 18 years of age) was taken from the female sex partners of PWID followed by pre-test counselling according to the national HIV testing guidelines [14]. Written assent was taken from parents or guardians of the participants. The assent was read out to the participants and the parents/guardians, and their signature or left thumb print impression was taken on the survey form. After pre-test counselling, HIV test was done by OraQuick HIV test kit using Oral Fluid [15]. Thereafter, a face to face interview was held with the participants using a semi-structured questionnaire. All questions were asked in Bangla and interviewers were trained thoroughly. At the end of training sessions, all procedures related to the study such as

obtaining written informed consent/assent, counselling (per and post), HIV testing using Ora-Quick HIV test kit, and risk behaviour interview was piloted outside of the Dhaka city. Experience from the field testing was used to fine-tune the quality of work and the questionnaire of measuring risk behaviours and vulnerabilities of the participants to enhance the validity of data and smooth running of the study. During the training, the importance of confidentiality and respect to the HIV positive PWID and their female sex partners was also emphasised. All interviews and HIV testing were held at Comprehensive Drop-in-Centres (DICs)/Drop-in-Centres (DICs) of the harm reduction program if the participant was able to come or at the residence of the participant where privacy and confidentiality could be maintained. HIV was tested using the OraQuick Rapid HIV Antibody test kit (OraSure Technologies, Inc. Bethlehem, PA 18015, USA) performed on Oral Fluid [15]. From those who's test result was found reactive with the OraQuick, 5ml blood was taken for confirmatory test using serum based rapid HIV testing by three WHO-recommended rapid test kits at the Virology Laboratory in Dhaka following the national HIV testing algorithm [14]. While collecting blood samples, all aseptic precautions were maintained and sharps were disposed following standard universal precautions [16]. If any of the serum-based WHO-recommended rapid test result was HIV negative, then the sample was confirmed by Line Immune Assay (Western blot). After the HIV confirmatory test, the result was shared with the doctor/counsellor of CDIC/DIC of the implementing agencies for conducting post-test counselling and taking measures to initiate HIV treatment, care and support (TCS) services.

## Timeline

Quantitative data were collected from June to November 2019 followed by data entry, analysis and report writing.

## Definition of risk behaviours and vulnerabilities

In this manuscript, HIV risk behaviours refer to 1) drugs and injection-related risk behaviours and 2) sexual risk behaviours with male sex partners such as: husband, clients, and non-transactional male sex partners (boyfriends/relatives/neighbours). Vulnerabilities refer to 1) treatment of and knowledge of STIs and HIV; and 2) uptake of routine HIV testing and HIV prevention services by the NGOs; and 3) Physical/sexual violence and other vulnerabilities.

## Definition of the comprehensive knowledge of HIV and measuring the progress of HIV testing services

The comprehensive knowledge of HIV was defined according to the UNAIDS [17] who simultaneously answered to five questions correctly such as, 1) "can people reduce their risk of HIV by using a condom correctly and consistently in any type of sex?" (correct answer is 'yes'), 2) "can people reduce their risk of HIV by avoiding sex with multiple partners?" (correct answer is 'yes'), 3) "can a person get HIV through mosquito bite?" (correct answer is 'no'), 4) "can a person get HIV by sharing a meal with someone who is HIV infected?" (correct answer is 'no'), and 5) "can you tell by looking at someone whether s/he is infected with HIV?" (correct answer is 'no'). Progress of the HIV testing was also assessed according to the UNAIDS definition [17] such as, 1) "have you been tested for HIV in the last 12 months?" and 2) "if yes, I don't want to know the results, but did you receive the results of that test?"

## Data management and analysis

The laboratory data were entered by Excel for each participant with a unique key identification number. The HIV risk behaviours and vulnerability data were entered twice by Epi-Info (Version 3.5) with range and consistency checks. Data were further cleaned by Excel before analysis. The categorical variables were described in terms of percentage points and numeric variables by medians along with inter-quartile range (IQR). In this manuscript all participants were combined into one group who represent female sex partners of the HIV positive PWID

**Table 1. Socio-demographic characteristics.**

| Indicators | N = 227 |
|---|---|
| | % (95%CI) |
| Age (in years) | |
| 15–19 | 6.6 (4.0–10.7) |
| 20–24 | 9.7 (6.4–14.3) |
| 25–34 | 35.7 (29.7–42.2) |
| 35–49 | 39.2 (33.0–45.8) |
| >49 | 8.8 (5.7–13.3) |
| Median (IQR) | 34.0 (27.0–41.0) |
| Years of schooling (in years) | |
| No education | 53.7 (47.2–60.2) |
| 1–5 | 32.2 (26.4–38.6) |
| 6–10 | 13.2 (9.4–18.3) |
| >10 | 0.9 (0.2–3.5) |
| Median (IQR) | 0 (0–4.0) |
| Income in the last month (in USD)§ | |
| Median (IQR) | 42 (0–85) |
| Main sources of income in the last month | |
| House wife | 30.0 (24.3–36.3) |
| Grocery shop/small enterprise | 10.1 (6.8–14.8) |
| Services | 10.6 (7.2–15.3) |
| Work at house-holds | 11.9 (8.3–16.8) |
| Sex work | 18.1 (13.6–23.7) |
| Garbage picker (Tokai) | 11.0 (7.5–15.8) |
| Day labour | 5.7 (3.3–9.6) |
| Beggar | 2.6 (1.2–5.8) |
| Current marital status | |
| Married | 82.4 (76.8–86.8) |
| Unmarried | 1.3 (0.4–4.0) |
| Divorced/Separated/Widowed | 15.4 (11.3–20.8) |
| Cohabitation | 0.9 (0.2–3.5) |
| Age at the first marriage (Denominator is who were ever married) | N = 224 |
| Median (IQR) | 15.0 (13.0–18.0) |
| Age at the first sex act | |
| Median (IQR) | 14.0 (12.0–16.0) |
| Currently live on the streets | 18.9 (14.3–24.6) |

§1 USD = 94.7 BDT

Note: IQR refers to inter quartile range

in Dhaka city. Data were analysed using Stata (version 13) for 227 participants among whom (136 were wives, 66 FSW and 25 non-transactional female sex partners (girl-friends/relatives/ neighbours) were combined into one group). Stata was also used to calculate 95% confidence interval (CI) for all categorical variables.

### Ethical assurance

The study proposal was reviewed and approved by the Institutional Review Board, which consists of the research review committee (RRC) and ethical review committee (ERC).

### Results

The target sample size in this study was 372 and finally, 227 (61.0%) were achieved. We faced challenges in interviewing FSWs and other female sex partners that have been mentioned in the challenges section. The study findings have been described below.

**Table 2. Drugs and injection related risk behaviours.**

| Indicators | N = 227 |
|---|---|
| | **(Unless otherwise stated)** |
| | **% (95% CI)** |
| Taking any kind of drugs in the last year | 24.7 (19.5–30.7) |
| Type of drugs taken in the last year | |
| (Denominator is who had taken drugs in the | N = 56 |
| last year)* | |
| Sleeping pill | 69.6 (56.0–80.5) |
| Cannabis | 58.9 (45.3–71.3) |
| Alcohol | 32.1 (21.0–45.8) |
| Phensidyl | 5.4 (1.7–15.8) |
| Heroin | 14.3 (7.1–26.5) |
| Buprenorphine/Pethidine (Injection) | 50.0 (36.8–63.2) |
| Methamphetamine (Yaba) | 78.6 (65.5–87.6) |
| andy/ Shoe gums | 7.1 (2.6–18.0) |
| Cocktail | 1.8 (0.2–12.3) |
| Duration of taking any kind of drugs (in | |
| years) (Denominator is who had taken any | N = 56 |
| kind of drugs in the last year) | |
| Median (IQR) | 11.0 (6.0–20.0) |
| Injected drugs in the last year | 12.3 (8.6–17.3) |
| Injected drugs in the last week | 9.3 (6.1–13.8) |
| Injections shared with husband/other males/friends in the last week (Denominator is who injected in the last week) | N = 21 |
| Always | 14.3 (4.2–38.7) |
| Sometimes | 38.1 (19.1–61.7) |
| Never | 47.6 (26.3–69.8) |

*Multiple responses

Note: IQR refers to inter quartile range

**Table 3. Sexual risk behaviours with male sex partners.**

| Indicators | N = 227 |
|---|---|
| | **(Unless otherwise stated)** |
| | **% (95%CI)** |
| *Sex with husband* | |
| Had sex with HIV positive husband in the last | N = 222* |
| year (Denominator is who were ever married) | 77.0 (71.0–82.1) |
| Frequency of condom use during vaginal sex | |
| acts with the husband in the last year | |
| (Denominator is who had sex with the | N = 171 |
| husband in the last year) | |
| Always | 18.7 (13.5–25.3) |
| Sometimes | 24.6 (18.6–31.6) |
| Never | 56.7 (49.1–64.0) |
| Used condom in the last vaginal sex act with | N = 171 |
| the husband in the last year (Denominator is | |
| who had sex with the husband in the last year) | 29.2 (22.9–36.6) |
| *Sex with clients (professional female sex workers)* | |
| Sold sex in the last year | 29.1 (23.5–35.4) |
| Frequency of condom use during vaginal sex | |
| acts with the male clients in the last year | |
| (Denominator is who sold sex to the male | |
| clients in the last year) | N = 66 |
| Always | |
| Sometimes | 15.2 (8.2–26.2) |
| Never | 66.7 (54.3–77.1) |
| | 18.2 (10.5–29.6) |
| Used condom in the last vaginal sex act with a | N = 66 |
| male client in the last year (Denominator is who sold sex to the male clients in the last year) | 43.9 (32.3–56.3) |
| *Sex with non-transactional partners (boy-friends/relatives/neighbours)* | |
| Had sex with the non-transactional male sex partners in the last year | 19.4 (14.7–25.1) |
| Frequency of condom use with the non- | |
| transactional male sex partners in the last year | |
| (Denominator is who had sex with the non- | |
| transactional male sex partners in the last | N = 44 |
| year) | |
| Always | 6.8 (2.1–19.9) |
| Sometimes | 22.7 (12.4–38.0) |
| Never | 70.5 (54.9–82.4) |
| Used condom in the last vaginal sex act with a | N = 44 |
| non-transactional male sex partner in the last | |
| year (Denominator is who had sex with non-transactional male sex partners in the last year) | 22.7 (12.4–38.0) |

*Two observations were missing

## Socio-demographics

The majority of the participants (74.9%) were 25–49 years old with median age of 34.0 years (IQR: 27.0–41.0) (Table 1). A little more than half of the participants had no education. The median income in the last month was approximately 42 USD (IQR: 0–85) (1 USD = 94.7 Taka, according to Bangladesh Bank, August 06, 2022) [18]. Mostly (82.4%, 95% CI: 76.8–86.8) were currently married. The median age of participants at the time of marriage was 15.0 years (IQR: 13.0–18.0). The median age at the first sex act was 14.0 years (IQR: 12.0–16.0). At the same time, 18.9% (95% CI: 14.3–24.6) lived on the streets.

## Prevalence of HIV

The prevalence of HIV was 16.7% (38 out of 227, 95% CI: 12.4–22.2); 31 were (13.7%, 95% CI: 9.5–18.8) detected previously through the routine HIV testing by the NGOs and 7 (3.1%, 95% CI: 1.2–6.2) were newly detected by this study. Of the 38 HIV positive participants, 33 (86.8%, 95% CI: 71.9–95.6) were in the reproductive age group 15–49 years. Further analysis of data showed that of the seven newly identified HIV cases, two were aged 15–24, three were aged 25–34 and two were aged 35–49 years old. Among those who were currently married, 8 participants were pregnant at the time of survey; of them no one was found HIV positive. All previously identified HIV positive cases were on the anti-retroviral treatment (ART). Of the 7 newly identified HIV positive cases, 4 cases were enrolled into the ART and 3 were still not traced/lost to follow-up. Later, repeated initiatives were taken by the implementing agency to find out the participants who were traceless but they were not found.

## HIV risk behaviours

**Drugs and injection related risk behaviours.** One-fourth (24.7%, 95% CI: 19.5–30.7) of the participants reported had taken any kind of drugs in the last year (Table 2). Among them, 78.6% (95% CI: 65.5–87.6) reported Methamphetamine (locally known as Yaba) as the most frequent drugs followed by 69.6% (95% CI: 56.0–80.5) sleeping pills (such as, Noctin, Enoktin, Triptin, Sedil, Milam, Tanodil, etc), 58.9% (95% CI: 45.3–71.3) cannabis and 50% (95% CI: 36.8–63.2) injecting drugs. The median duration of taking any kind of drugs was reported at 11.0 years (IQR: 6.0–20.0). Among those who injected drugs in the last week, 14.3% (95% CI: 4.2–38.7) reported shared needles/syringes always and 38.1% (95% CI: 19.1–61.7) reported sometimes with husband/other male injecting drug users.

**Sexual risk behaviours with the male sex partners.** Among those who were currently married at the time of survey, sex with HIV positive husband in the last year was reported by 77% (95% CI: 71.0–82.1) of the participants (Table 3). In the last year, only 18.7% (95% CI: 13.5–25.3) reported consistent condom use during all sex acts and only 29.2% (95% CI: 22.9–36.6) used condom in the last sex act. Of 227 participants, 66 (29.1%, 95% CI: 23.3–35.4) were professional female sex workers (FSW). In the last year, consistent use of condoms was reported by only 15.2% (95% CI: 8.2–26.2) of the participants and the last time condom use in the last sex act with a client was reported by 43.9% (95% CI: 32.3–56.3). Approximately, one in every five participants (19.4%, 95% CI: 14.7–25.1) had sex with non-transactional male sex partners (except husband and clients) in the last year. Consistent use of condoms was reported by only 6.8% (95% CI: 2.1–19.9) and the last time condom use in the last sex act was reported by 22.7% (95% CI: 12.4–38.0) of the participants in the last year.

## Vulnerability to HIV

**Treatment of and knowledge of the STIs and HIV.** Three in every four participants (75.3%, 95% CI: 69.2–80.8) did not have any knowledge on symptoms of sexually transmitted infections (STIs) (Table 4). Self-reported symptoms of any STIs, such as: i) painful or smelly discharge; or ii) lower abdominal pain which was not associated with menstruation, or a stomach upset; or iii) genital warts, sores, or ulcers, were reported by 26% (95% CI: 20.7–32.1) of the participants. A little more than half of the participants received treatment for STIs in the last year. Only 19.4% (95% CI: 14.7–25.1) had comprehensive knowledge of HIV.

## Uptake of the routine HIV testing and HIV prevention services providing by the NGOs

Approximately two-third of the study participants (67.4%, 95% CI: 61.0–73.2) reported that they knew about HIV testing through the NGOs (Table 5). Three in every five (61.7%, 95% CI: 55.1–67.8) study participants reported ever testing for HIV and 38.3% (95% CI: 32.2–44.9) mentioned never tested for HIV in their lifetime. Among those who never tested for HIV, participants mostly (87.4%, 95% CI: 78.4–92.9) mentioned 'no one told me to test'. In the last year, as a routine HIV prevention services by the PWID Drop-in-Centres (DICs), 42.7% (95%

**Table 4. Treatment of and knowledge of the STIs and HIV.**

| Indicators | N = 227 |
|---|---|
| | **(Unless otherwise stated)** |
| | **% (95%CI)** |
| Knowledge about STI symptoms* | |
| Don't know | 75.3 (69.2–80.8) |
| Vaginal discharge | 9.7 (6.4–14.3) |
| Smelly discharge | 8.4 (5.4–12.8) |
| Genital ulcers/sores | 15.4 (11.3–20.8) |
| Lower abdominal pain | 7.5 (4.7–11.8) |
| Itching | 8.8 (5.7–13.3) |
| Reported at least one STI symptom [i) painful or smelly discharge, or ii) lower abdominal pain which was not associated with menstruation, or a stomach upset, or iii) genital warts, sores, or ulcers] in the last year | 26.0 (20.7–32.1) |
| Received treatment for any STI in the last | N = 59 |
| year (Denominator is who had any STI | |
| symptoms in the last year) | 57.6 (44.4–69.8) |
| Choice of the STI treatment in the last year | N = 34 |
| (Denominator is who reported STI symptoms | |
| in the last year) | |
| Govt. Hospital | 17.6 (7.8–35.2) |
| Pharmacy | 11.8 (4.3–28.6) |
| Private doctor | 5.9 (1.4–22.0) |
| NGO clinic | 50.0 (33.0–67.0) |
| Canvasser/Traditional healer | 11.8 (4.3–28.6) |
| Self-medication | 2.9 (0.4–19.8) |
| Ever heard about HIV/AIDS | 93.0 (88.8–95.7) |
| Had comprehensive knowledge of HIV | 19.4 (14.7–25.1) |

**Table 5. Uptake of the routine HIV testing and HIV prevention services by the NGOs.**

| Indicators | N = 227 |
|---|---|
| | **(Unless otherwise stated)** |
| | **% (95%CI)** |
| *HIV testing* | |
| Knew where HIV can be tested confidentially by the NGOs | 67.4 (61.0–73.2) |
| Tested for HIV by the NGOs | |
| Ever | 61.7 (55.1–67.8) |
| Never | 38.3 (32.2–44.9) |
| Reasons for never tested for HIV | N = 87 |
| (Denominator is who never tested for HIV) | |
| No one told me to test | 87.4 (78.4–92.9) |
| Don't think I am at risk | 12.6 (7.1–21.6) |
| Husband is well | 1.1 (0.2–8.0) |
| Progress of the HIV testing services (Received the HIV testing and counselling in the last year and knew the result) | 42.7 (36.4–49.3) |
| *HIV prevention services* | |
| Received any HIV/AIDS prevention services | |
| Within a year | 64.3 (57.8–70.3) |
| More than year ago | 3.1 (1.5–6.4) |
| Never | 32.6 (26.8–39.0) |
| Type of services received in the last year | |
| (Denominator is who received any services in | N = 146 |
| the last year) | |
| Attended education program | 22.6 (16.5–30.2) |
| Attended HTS | 90.4 (84.4–94.3) |
| Received condom | 36.3 (28.8–44.5) |
| Received treatment for the STIs | 15.1 (10.1–21.9) |
| Received treatment for the general health | 29.5 (22.6–37.4) |
| Received ANC services | 2.7 (1.0–7.1) |
| Sleeping/Shower/Eating/Nutrition | 30.1 (23.2–38.2) |
| Watching TV/Playing games | 16.4 (11.2–23.4) |
| Received Opioid Substitution Therapy services (OST) | 5.5 (2.7–10.7) |

CI: 36.4–49.3) received HIV testing services from the PWID DICs as a routine HIV prevention service and knew the result.

In the last one year, 64.3% (95% CI:57.8–70.3) reported receiving any HIV prevention services from the CDICs/DICs (Table 5). On the other hand, approximately one-third of the participants never received any HIV prevention services. Among those who had received services in the last year, mostly mentioned received HIV testing and services (HTS) (90.4%, 95% CI: 84.4–94.3). Of those who received services in the last year, only 36.3% (95% CI: 28.8–44.5) of the participants mentioned received condoms.

## Physical and sexual violence, and other vulnerabilities

The data on physical and sexual violence and other vulnerabilities showed that in the last year, 45.9% (95% CI: 39.3–52.6) participants reported been beaten and 20.2% (95% CI: 15.3–26.1) reported been raped (Table 6). Most of the episodes of beating and rape were perpetuated by the husbands (61–67%). Further analysis of data showed that out of 136 wives, 40 (29.4%, 95%

**Table 6. Physical and sexual violence, and other vulnerabilities.**

| Indicators | N = 227 |
|---|---|
| | **(Unless otherwise stated)** |
| | **% (95%CI)** |
| Reported been beaten in the last year | N = 218[§] |
| | 45.9 (39.3–52.6) |
| Beating perpetuated by[*] | |
| (Denominator is who reported been beaten in the last year) | N = 100 |
| Men in uniform | 7.0 (3.3–14.1) |
| Mastan (Hoodlums) | 6.0 (2.7–12.9) |
| New sex partner | 5.0 (2.1–11.6) |
| Regular sex partner | 5.0 (2.1–11.6) |
| Local people | 13.0 (7.6–21.3) |
| Non- transactional male sex partner | 7.0 (3.3–14.1) |
| Husband | 67.0 (57.1–75.6) |
| In-laws | 2.0 (0.5–7.8) |
| Parents/Brother/Sister | 5.0 (2.1–11.6) |
| Son/Son's wife | 1.0 (0.1–7.0) |
| Reported been raped in the last year | N = 218 |
| | 20.2 (15.3–26.1) |
| Rape perpetuated by[*] | |
| (Denominator is who reported been raped in the last year) | N = 44 |
| Men in uniform | 11.4 (4.6–25.3) |
| Mastan (Hoodlums) | 22.7 (12.4–38.0) |
| New sex partner | 13.6 (6.1–27.9) |
| Regular sex partner | 6.8 (2.1–19.9) |
| Local people | 25.0 (14.1–40.4) |
| Non- transactional male sex partner | 9.1 (3.3–22.6) |
| Husband | 61.4 (45.8–74.9) |
| Reported been beaten or raped in the last year | N = 218 |
| | 50.5 (43.8–57.1) |
| Knew that the husband injects drugs | N = 222 |
| (Denominator is who were ever married) | 44.6 (38.1–51.2) |
| Had new/regular clients who inject drugs | N = 66 |
| (Denominator is who were FSW) | 56.1 (43.6–67.8) |
| Had non-transactional male sex partners who | N = 44 |
| inject drugs (Denominator is who had sex with | |
| non-transactional male sex partners in the last year) | 54.5 (39.3–69.0) |

[§]9 observations were missing

[*]Multiple responses

CI: 22.3–37.7) reported been beaten within the last year. Of 40 wives, 34 (85.0%, 95% CI: 70.1–93.2) reported that they were beaten by their husband who is a HIV positive PWID. Of 136 wives, 13 (9.6%, 95% CI: 5.6–15.8) reported that they have been raped in the last year. Of them, everyone mentioned that they were raped by their husband who is a HIV positive PWID. FSWs and other female sex partners also reported that they were been beaten and raped by their husband who were non-PWID and the status of HIV infection of these husbands were not known. Among those who were ever married, 44.6% (95% CI: 38.1–51.2) knew that the

husband injects drugs. Of those who sold sex (professional sex workers) in the last year, 56.1% (95% CI: 43.6–67.8) knew that their clients inject drugs and of those who had sex with non-transactional male sex partners in the last year, 54.5% (95% CI: 39.3–69.0) knew that their non-transactional male sex partners inject drugs.

## Discussion

The findings of this study revealed the characteristics, prevalence of HIV, risk behaviours and vulnerabilities among female sex partners of the HIV positive PWID in Dhaka city. This is the first of its kind of study in Bangladesh among these population groups, from which comprehensive quantitative data are available that can be useful to re-design or strengthen HIV prevention services for the studied population groups in Dhaka city.

The prevalence of HIV among the study participants was 16.7% which crossed the concentrated epidemic level 5% [19] and coincides (2.5%-45.0%) with the data from similar groups in Pakistan, India, Kazakhstan, Vietnam, Iran, Russia, Kyrgyzstan and Malaysia [3–10, 20]. On the other hand, the prevalence of HIV among the PWID (who were the husbands or male sex partners of this study participants) in Dhaka city was 22% in [1]. Studies conducted in Vietnam and Pakistan showed that the PWID male sex partners were one of the major sources of HIV infection among their female sex partners [21, 22]. Secondary analysis of data among 400 wives of the PWID in India showed that the wives were 17.9 times more likely to be infected with HIV if the PWID husbands were HIV infected (p<0.001) [20]. The sources of the HIV infection among the participants in this study can also be explained by the HIV case reporting data. The HIV case reporting data in Bangladesh showed that in 2020, 534 new HIV cases were detected, and of which 21% were female, 76% were male and 3% were hijra (transgender) [23]. The HIV case reporting data also showed that of 534 new HIV infections, 24.7% were PWID. Similar data from Iran in 2010 showed that 8.7% of 9,136 new HIV cases were detected among women, and of which, 76% of the women were infected by their husbands who were primarily PWID [24]. In this study, a large number of the participants (83.3%) were still HIV negative despite having sex with a HIV positive PWID. This may be because the HIV positive PWID are on the ART for a long period of time and they are virally suppressed therefore, are not transmitting HIV to their female sex partners [25]. However, a representative survey of the HIV viral load testing among the HIV positive PWID in Dhaka city is needed to be conducted to find out what percentage of the PWID are virally suppressed so that transmission of HIV to their female sex partners has become limited [26]. In this study, of a total of the 38 HIV positive participants, 31 (13.6%) were old cases identified before by the NGOs and 7 (3.1%) were newly identified in this study. Identification of the new HIV cases in this study was similar with a study conducted among 364 female sex partners of the PWID in Kazakhstan that detected only 19 participants (5.2%) who were newly infected with HIV [6].

Taking any kind of drugs was a common risk behaviour among the female sex partners of the PWID in other countries and we also found the same in our study which more closely reflects with estimates from similar population groups (16%-58%) in India, Iran, Vietnam and Pakistan [7, 20, 27, 28]. Among those who had taken drugs in the last year, Yaba and sleeping pills (such as, Noctin, Enoktin, Triptin, Sedil, Milam, Tanodil, etc) were the most popular. A study conducted among the HIV positive PWID in Vietnam showed that the use of methamphetamine may increase the likelihood of the HIV transmission from a HIV positive PWID to non-injecting female sexual partners due to its potential to decrease the ART adherence, which would consequently weaken the immune system and increase viral load [29]. Of 21 study participants who injected drugs during the last week, 11 (52.4%) reported shared needles/syringes always or sometimes with their husbands or other males that signals in addition to the sexual

transmission, HIV infection may have been transmitted to them by sharing contaminated needles/syringes as was reported in a study among similar population in India [30]. The low rate of condom use (both during the last sex act and consistently) demonstrated that the female sex partners who were still HIV negative were at a great risk of HIV infection. Similar situation was observed among the female sex partners of the HIV positive PWID in Pakistan [4].

In this study, knowledge of the STI infection and treatment-seeking behaviour was poor. Similar findings were observed (9.9%-44.7%) among the women of the HIV positive PWID in India and Pakistan [3, 4]. Seeking treatment for the STIs was low by the participants in this study that indicates the unmet need for the STIs prevention and treatment for these population groups is urgent to reduce the risk of the HIV infection. The comprehensive knowledge of HIV was found very low (19.4%) among the study participants compared to 42.5% among similar participants in Pakistan [4]. The comprehensive knowledge of HIV was found significantly lower (p<0.001 for both comparisons) among the participants in this study compared to 41.5% general female population in Bangladesh [31] and 68% in Pakistan [32], respectively. A study conducted among the 15–49 ever-married women in Pakistan showed that the comprehensive knowledge of HIV was 1.25 times higher among the women who were exposed to the mass-media than those who were not exposed to the mass-media [32] that suggests providing the knowledge on HIV is urgently needed by the mass-media such as radio/TV/Social media/ News paper in our country.

In this study, a considerable percentage of the participants were never tested for HIV due to lack of the designated field staff and in some DICs the participants live far away. Out of 227 sampled, 38.3% (n = 87) never tested for HIV that indicates they were hard to reach by the current HIV prevention services. Of them, mostly (87.4%) mentioned they were not aware of the HIV testing by the NGOs that demands immediate action. A study conducted in Vietnam among the female sex partners of the PWID showed that the HIV testing went up to 65% after 2 years of interventions from 37% at the baseline [33].

Findings in this study highlight that the services for these participants were available mostly for HIV testing and enrolled them into the ART when they were identified as HIV positive. At the same time, a promising result is that of the previously diagnosed HIV positive participants (n = 31), everyone was on the ART. A few of the participants mentioned that they received condoms (36.3%) or attended educational programs (22.6%) that signalled condom promotion was inadequate. On the other hand, approximately one-third of the participants never received any services due to the lack of the designated field staff. All these findings highlight that the HIV prevention program for these population groups need to be reviewed and then strengthened.

Targeted HIV prevention programmes have been implemented among key populations (KPs) who are at risk of HIV in Bangladesh since 1995 [34] and the harm reduction program for the people who inject drugs (PWID) started in Dhaka in 1998 [35]. Currently, the services are being provided by the NGOs through static CDICs/DICs and outreach workers in the drug-taking spots where PWID (both male and female) assemble to take drugs. The services include distribution of sterile needles/syringes, distribution of condoms and behaviour change communication (BCC) materials, CDIC/DIC based clinical services for managing abscess or injection related infections and STIs, HIV testing services (HTS), along with motivational education (one-to-one and in a group), provision of TB treatment, anti-retroviral treatment (ART) as per need and opioid substitution therapy (OST) [36]. Females who inject drugs (FWID) with suspected STIs found during syndromic diagnosis at the DICs are referred to the DICs of FSW for the STI management. They are also sent to the FSW DICs for other sexual and reproductive health services (SRH). There is an option for testing female partners of the HIV positive PWID at CDICs/DICs and if needed ART is provided to them. In addition,

female sex partners of the HIV positive PWID are linked to the tertiary hospitals for pregnancy care, breastfeeding counselling and early infant diagnosis. However, it may be mentioned that testing HIV for female sex partners of the HIV positive PWID are being conducted on a limited scale. When our study was in a process of planning during February 2019, in total, till December 2018, only 19 wives of the HIV positive PWID were identified as HIV positive (personal communication with CARE Bangladesh).

## Strengths, weaknesses and challenges

This study has several strengths, weaknesses and challenges. The first and foremost strength was that, this study demonstrated for the first time in Bangladesh the success of utilizing a network of the HIV positive PWID to access their female sex partners in identifying new cases of HIV and exploring detail information on socio-demographics, HIV risk behaviours and vulnerabilities who were otherwise difficult to reach by the current HIV prevention services alone [37]. Another strength of this study was the feasibility of using OraQuick rapid testing as a way of identifying new HIV cases among these participants. The limitations were, 1) behavioural information was collected through a face-to-face interview therefore, responses might have some social desirability bias [38], and 2) some information on drugs and sexual risk behaviours might suffer from recall bias [39].

During data collection we faced some challenges. Firstly, it was very challenging to take consent from the HIV positive PWID to get access to their female sex partners. Initially, they were very afraid that the study team would disclose his HIV status to his wife or his other female sex partner. When the study team spent some time explaining the objectives and benefits of this study, they agreed. Secondly, non-drug user female sex partners did not feel comfortable coming to the CDICs/DICs because at that time other injecting drug users were there and the place was crowded. Thirdly, some female sex partners lived far away from the CDICs/DICs. In that case, the team members went to their home and completed HIV testing and interview. The HIV positive PWID were not been able to identify the FSWs properly because the sex act was performed at night in many places with multiple FSWs during a period of time. Also, other female sex partners were reluctant to come to the CDICs/DICs or did not allow the team members to go to them due to the fear of being disclosed of her affair with an injecting drug user to the society/family. Due to these reasons, data collection against the sample sizes for FSWs and girl-friends/relatives/neighbours were not full-filled.

## Recommendations

Based on the findings and discussions, we would like to recommend:

1. The HIV prevention programs targeted to the female sex partners of the HIV positive PWID in Dhaka city need to be reviewed to find out strengths and weaknesses, and need more attention from the relevant stake holders.

2. Existing HIV prevention services for the HIV positive PWID should also include components for their female sex partners [40, 41] such as condom distribution and promotion, and educational program.

3. Counselling of couples need to be ensured. A study conducted among 4,612 female regular sex partners of the PWID in Bangladesh, Bhutan, India, Nepal and Sri Lanka showed that women who were approached and counselled by the service providers with information on HIV/AIDS, were 1.7 times (95% CI: 1.4–2.0) more likely to use condoms in the last sex act with their PWID sex partners [41]. Therefore, the HIV positive PWID can be encouraged to bring their female sex partners into a CDIC/DIC for couple counselling in a particular

time slot [30]. Counselling of influential family members such as father/mother/in-laws is also needed to make an enabling environment to provide services to these population groups [42].

4. Female sex partners of the HIV positive PWID who are still HIV negative need to be tested preferably six monthly or at least once a year [14] for early detection of HIV and early initiation to the ART that can play an important role in controlling the epidemic.

5. Initiatives needed to be taken to improve the knowledge related to HIV/STI among the study participants by developing information materials that are easily understandable and preferably delivered by female workers/staff [4]. Studies in the Northern India showed that due to the lack of female workers in the DICs, PWID do not want to bring their wives to take services [30]

6. Utilization of treatment for the STIs can be further increased by appointing female doctors/ Medical Assistants from the nearby DICs where the participants live so that she can be followed-up to ensure access and utilization of services routinely [30]

7. A representative survey of the HIV viral load testing among the HIV positive PWID in Dhaka city is urgently needed to be conducted to find out what percentage of them are virally suppressed that limits the transmission of HIV to their female sex partners [26]

## Conclusion

The findings of this manuscript highlight that the needs of the female sex partners of the HIV positive PWID have never been effectively identified and met. Therefore, it is urgently necessary to consider the high-risk behaviours, vulnerabilities, and challenges in designing or to strengthen targeted interventions for female sex partners of the HIV positive PWID in Dhaka city to ensure equality in access and utilization of services.

## Supporting information

**S1 File.**
(PDF)

**S2 File.**
(PDF)

## Acknowledgments

First of all, we would like to express our gratitude to the HIV positive PWID in Dhaka city for providing their consent allowing us to interview their female sex partners in this study. We also thank female participants who provided blood and giving their time in responding to questions. Without their active participation the survey would not have been possible and they are therefore acknowledged gratefully. We are thankful to the Save the Children International Bangladesh and CARE Bangladesh allowing us using their comprehensive drop-in-centres (CDICs)/ drop-in-centres (DICs) for data collection venue. We also thank all staff members in the CDICs/DICs for their cordial co-operation in recruiting participants. We thank Mr. Abu Taher for his constant supervision in data collection. We are grateful to AIDS/STD programme (ASP) for overall support, coordination and monitoring of the field activities.

## Author Contributions

**Conceptualization:** Md. Masud Reza, A. K. M. Masud Rana, Sharful Islam Khan.

**Data curation:** Md. Masud Reza.

**Formal analysis:** Md. Masud Reza.

**Methodology:** Md. Masud Reza.

**Project administration:** Md. Masud Reza, Sharful Islam Khan.

**Resources:** Md. Masud Reza, Sharful Islam Khan.

**Supervision:** Md. Masud Reza, A. K. M. Masud Rana, Mohammad Niaz Morshed Khan, Md. Safiullah Sarker, Sujan Chowdhury, Mohammad Ezazul Islam Chowdhury, Md. Abu Taher, Sharful Islam Khan.

**Validation:** Md. Masud Reza, A. K. M. Masud Rana, Sharful Islam Khan.

**Writing – original draft:** Md. Masud Reza, A. K. M. Masud Rana.

**Writing – review & editing:** Md. Masud Reza, A. K. M. Masud Rana, Mohammad Niaz Morshed Khan, Md. Safiullah Sarker, Sujan Chowdhury, Md. Ziya Uddin, Lima Rahman, Mohammad Ezazul Islam Chowdhury, Md. Abu Taher, Sharful Islam Khan.

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
