## [Decision Letter · Decision Letter 0]

2 Jan 2023

PONE-D-22-26066Prevalence of HIV, risk behaviours and vulnerabilities of female sex partners of HIV positive Men Who Inject Drugs (MWID) in Dhaka city, Bangladesh.PLOS ONE

Dear Dr. Khan,

Thank you for submitting your manuscript to PLOS ONE. After careful consideration, we feel that it has merit but does not fully meet PLOS ONE’s publication criteria as it currently stands. Therefore, we invite you to submit a revised version of the manuscript that addresses the points raised during the review process.

As you can see the reviewers have raised concerns related to methodology, incomplete information on statistical analysis and poorly written discussion. Additionally, they have suggested improvements in the language of the paper and making it short and to-the-point wherever possible. 

We look forward to receiving your revised manuscript.

Kind regards,

Syed Hani Abidi

Academic Editor

PLOS ONE

Journal Requirements:

" ext-link-type="uri" xlink:type="simple">https://journals.plos.org/plosone/s/file?id=ba62/PLOSOne_formatting_sample_title_authors_affiliations.pdf"

3. You indicated that you had ethical approval for your study. In your Methods section, please ensure you have also stated whether you obtained consent from parents or guardians of the minors included in the study or whether the research ethics committee or IRB specifically waived the need for their consent.

"Without their active participation the survey would not have been possible and they are therefore acknowledged gratefully. We are thankful to the Save the Children Bangladesh and CARE Bangladesh allowing us using their Comprehensive Drop-in-centres (CDICs)/ Drop-in-centres (CDICs) for data collection venue. We also thank all staff members in the CDICs/DICs for their cordial co-operation in recruiting participants. We thank Mr. Abu Taher for his constant supervision in data collection. We are grateful to AIDS/STD Programme (ASP) for overall support, coordination and monitoring of field activities. 

This survey was funded by The Global Fund to Fight AIDS, Tuberculosis and Malaria (GFATM), through the Grant ‘Expanding HIV/AIDS Prevention in Bangladesh’ with icddr,b. icddr,b acknowledges with gratitude the commitment of the Global Fund to its research efforts. icddr,b is also thankful to the Governments of Bangladesh, Canada, Sweden and the UK for providing core/unrestricted support to icddr,b."

"This survey was funded by The Global Fund to Fight AIDS, Tuberculosis and Malaria (GFATM), through the Grant ‘Expanding HIV/AIDS Prevention in Bangladesh’ under the terms of Grant Agreement NO. GR-01603 with icddr,b. icddr,b acknowledges with gratitude the commitment of the Global Fund to its research efforts. icddr,b is also thankful to the Governments of Bangladesh, Australia, Canada, Sweden and the UK for providing core/unrestricted support to icddr,b."

Reviewers' comments:

Reviewer's Responses to Questions

**Comments to the Author**

1. Is the manuscript technically sound, and do the data support the conclusions?

Reviewer #1: Yes

Reviewer #2: Partly

2. Has the statistical analysis been performed appropriately and rigorously? 

Reviewer #1: Yes

Reviewer #2: I Don't Know

3. Have the authors made all data underlying the findings in their manuscript fully available?

Reviewer #1: Yes

Reviewer #2: No

4. Is the manuscript presented in an intelligible fashion and written in standard English?

Reviewer #1: Yes

Reviewer #2: No

5. Review Comments to the Author

Reviewer #1: • Suggestion to revise the title: Prevalence of HIV, risk behaviours and vulnerabilities of female sexual partners of HIV positive people who inject drugs in Dhaka, Bangladesh

• From the outset, please be clear that this study was done on spouses of male injection drug users. Your paper is still valuable even if most (82.4%) of the study participants were spouses.

Abstract

• People who inject drugs (PWID) is more renown and globally used term.

• Please write IBBS for the first time in full.

• A more focused last sentence of the background section could be: This paper describes the HIV prevalence, risk behaviours, and vulnerabilities of PWIDs and their sexual partners.

• Please revise methods: This was a cross-sectional study conducted among 227 female sexual partners of HIV-positive MWID or PWID in Dhaka, Bangladesh, in 2019, adopting a take-all sampling technique.

• Please write STI in full for the first time.

• What does “receive” in this sentence mean: Forty three percent (95% CI: 36.4-49.3) received HIV testing services and knew the result during last year. Please rephrase if this was part of outreach through the DICs.

• One-third never received any services. Are these HIV prevention services?

• Please stick to the same pattern as you have done in previous sentences, which is to write in words instead of numbers after a full stop. 46% (95% CI: 39.3-52.6) reported been beaten and 20.2% (95% CI: 15.3-26.1) been raped during last year.

• Husbands? Most of the episodes of beating and rape were perpetuated by HIV-positive husbands (61-67%). If this is about spouses, then please revise the title.

• The conclusion does not match the findings. Please revise.

Introduction

• As mentioned earlier, PWID is a more common term used globally.

• Line 56: ..please revise to has risen from was risen.

• Lines 70-74 need to be revised. Please only mention the rationale and objective here.

Methods

• Please revise the methods in the following pattern: study type, population, sample size, sampling, and timeline of data collection. This is how the methods section should ideally start. The rest of the sections can follow.

• Is there a difference between a drop in centre and a comprehensive drop in centre?

Results

• The narration in the results section needs to be trimmed. It is far too long, and the reader gets lost. Mention key findings and refer to the tables.

• Please give serious thought to merging variables of tables 3-6 into one table.

• The mean age of 15.6 years; is the time of being married or the mean age of the respondent at the time of marriage?

• The median number can be deleted from the table.

• Line 187: this is the spouse/sexual partner of the drug user? Please clarify.

• It is a suggestion to highlight condom use in the last sexual act (22.7%) prominently, as this is far more important than consistent condom use in the last year, which has a high probability of recall bias.

Conclusion

• Please reduce the conclusion to two to three sentences.

Reviewer #2: Thanks for the review. The article describes prevalence and risk behaviours of partners of HIV positive men who inject drugs.

General comment: The language requires editing. e.g Professional sex workers instead of " selling sex" among others.

Comment 1:

"Four new cases are on ART and 3 are still no trace". Does it mean that they have been lost to follow up?

Comment 2: Please share response rate to questionnaires in first paragraph of results. Numbers keep changing in all tables and there is significant attrition of numbers while in text only percentages have been reported.

Comment 3: Would be best to report medians and IQRs throughout rather than reporting means and medians both.

Comment 4: How were 95% CI computed and what was the point estimate. There is no mention in methods regarding what statistical tests have been used.

Comment 5: Subheadings in discussion are not required

Comment 6: The discussion lacks appropriate comparisons with local and regional settings. Should elaborate why the uptake was not sufficient and how that was comparable to other countries. In some parts discussion seems like an elaborate write up of results and needs to be re-written as discussion especially the paragraphs on uptake and vulnerabilities.

6. PLOS authors have the option to publish the peer review history of their article (what does this mean?). If published, this will include your full peer review and any attached files.

Reviewer #1: **Yes: **Arshad Altaf

Reviewer #2: No

---

## [Author Response · Author response to Decision Letter 0]

4 Apr 2023

Thank you for giving us the opportunity to revise this manuscript. As per the respected reviewers' feedback, we have incorporated various revisions, which have been detailed in the Response to Reviewers letter. Please find the revisions indicated in track changes. We have also addressed the issue regarding the data availability statement.

---

## [Decision Letter · Decision Letter 1]

22 May 2023

Prevalence of HIV, risk behaviours and vulnerabilities of female sex partners of HIV positive People Who Inject Drugs (PWID) in Dhaka city, Bangladesh

PONE-D-22-26066R1

Dear Dr. Khan,

We’re pleased to inform you that your manuscript has been judged scientifically suitable for publication and will be formally accepted for publication once it meets all outstanding technical requirements.

Kind regards,

Syed Hani Abidi

Academic Editor

PLOS ONE

Additional Editor Comments (optional):

Reviewers' comments:

Reviewer's Responses to Questions

**Comments to the Author**

1. If the authors have adequately addressed your comments raised in a previous round of review and you feel that this manuscript is now acceptable for publication, you may indicate that here to bypass the “Comments to the Author” section, enter your conflict of interest statement in the “Confidential to Editor” section, and submit your "Accept" recommendation.

Reviewer #1: All comments have been addressed

Reviewer #3: All comments have been addressed

2. Is the manuscript technically sound, and do the data support the conclusions?

Reviewer #1: Yes

Reviewer #3: Yes

3. Has the statistical analysis been performed appropriately and rigorously? 

Reviewer #1: Yes

Reviewer #3: Yes

4. Have the authors made all data underlying the findings in their manuscript fully available?

Reviewer #1: Yes

Reviewer #3: Yes

5. Is the manuscript presented in an intelligible fashion and written in standard English?

Reviewer #1: Yes

Reviewer #3: No

6. Review Comments to the Author

Reviewer #1: Thank you for addressing the comments.

For this particular comment: • The median number can be deleted from the table.

RESPONSE: The second reviewer suggested to delete mean values and keep the median values.

In that case, I am confused what to do? You can keep median values.

Reviewer #3: Minor comment: Unnecessary use of the articles (a/the) found throughout the manuscript, please revise the manuscript accordingly, for example, the result section of the abstract says, “Mean age of”, should be “The mean age of”. Similarly, in the same section, the sentence read: “Condoms were used consistently with different male sex partners; only 6.8% to 18.7% during last year”; please re-write this sentence in standard English format. Numerous similar grammatical mistakes are found throughout the manuscript, please carefully read your manuscript, and address similar issues.

7. PLOS authors have the option to publish the peer review history of their article (what does this mean?). If published, this will include your full peer review and any attached files.

Reviewer #1: **Yes: **Arshad Altaf

Reviewer #3: No

---

## [Editor Report · Acceptance letter]

26 May 2023

PONE-D-22-26066R1 

Prevalence of HIV, risk behaviours and vulnerabilities of female sex partners of the HIV positive people who inject drugs (PWID) in Dhaka city, Bangladesh 

Dear Dr. Khan:

I'm pleased to inform you that your manuscript has been deemed suitable for publication in PLOS ONE. Congratulations! Your manuscript is now with our production department. 

Kind regards, 

on behalf of

Dr. Syed Hani Abidi 

Academic Editor

PLOS ONE